# Longitudinal Characterization of Immune Response in a Cohort of Children Hospitalized with Multisystem Inflammatory Syndrome

**DOI:** 10.3390/children10061069

**Published:** 2023-06-16

**Authors:** Laura Dotta, Daniele Moratto, Marco Cattalini, Sara Brambilla, Viviana Giustini, Antonella Meini, Maria Federica Girelli, Manuela Cortesi, Silviana Timpano, Anna Galvagni, Anna Viola, Beatrice Crotti, Alessandra Manerba, Giorgia Pierelli, Giulia Verzura, Federico Serana, Duilio Brugnoni, Emirena Garrafa, Francesca Ricci, Cesare Tomasi, Marco Chiarini, Raffaele Badolato

**Affiliations:** 1Department of Pediatrics, ASST Spedali Civili of Brescia, University of Brescia, 25123 Brescia, Italy; marco.cattalini@unibs.it (M.C.); antonella.meni@asst-spedalicivili.it (A.M.); mariafederica.girelli@asst-spedalicivili.it (M.F.G.); silviana.timpano@asst-spedalicivili.it (S.T.); raffaele.badolato@unibs.it (R.B.); 2Department of Clinical and Experimental Sciences, University of Brescia, 25123 Brescia, Italy; cesare.tomasi@live.com; 3Flow Cytometry Unit, Clinical Chemistry Laboratory, ASST Spedali Civili of Brescia, 25123 Brescia, Italy; daniele.moratto@asst-spedalicivili.it (D.M.); viviana.giustini@gmail.com (V.G.); galvagni.anna@gmail.com (A.G.); duilio.brugnoni@asst-spedalicivili.it (D.B.); marco.chiarini@asst-spedalicivili.it (M.C.); 4Pdiatric Cardiology Unit, ASST Spedali Civili of Brescia, 25123 Brescia, Italy; alessandra.manerba@asst-spedalicivili.it (A.M.);; 5Hematology Unit, Clinical Chemistry Laboratory, ASST Spedali Civili of Brescia, 25123 Brescia, Italy; federico.serana@asst-spedalicivili.it; 6Laboratory of Clinical Chemistry, ASST Spedali Civili of Brescia, 25123 Brescia, Italy; emirena.garrafa@unibs.it; 7Department of Molecular and Translational Medicine, University of Brescia, 25123 Brescia, Italy

**Keywords:** MIS-C, SARS-CoV-2, COVID, lymphopenia, monocytopenia, immunophenotype, multiparametric flow cytometry

## Abstract

Background: Multisystem Inflammatory Syndrome in Children (MIS-C) is a severe complication of SARS-CoV-2 infection caused by hyperactivation of the immune system. Methods: this is a retrospective analysis of clinical data, biochemical parameters, and immune cell subsets in 40 MIS-C patients from hospital admission to outpatient long-term follow-up. Results: MIS-C patients had elevated inflammatory markers, associated with T- and NK-cell lymphopenia, a profound depletion of dendritic cells, and altered monocyte phenotype at disease onset, while the subacute phase of the disease was characterized by a significant increase in T- and B-cell counts and a rapid decline in activated T cells and terminally differentiated B cells. Most of the immunological parameters returned to values close to the normal range during the remission phase (20–60 days after hospital admission). Nevertheless, we observed a significantly reduced ratio between recently generated and more differentiated CD8+ T- and B-cell subsets, which partially settled at longer-term follow-up determinations. Conclusions: The characterization of lymphocyte distribution in different phases of MIS-C may help to understand the course of diseases that are associated with dysregulated immune responses and to calibrate prompt and targeted treatments.

## 1. Introduction

Since April 2020, after the worldwide spread of the coronavirus SARS-CoV-2 infection, numerous reports have described a novel condition associated with coronavirus disease 2019 (COVID-19) and named Multisystem Inflammatory Syndrome in Children (MIS-C) or Pediatric Inflammatory Multisystem Syndrome temporarily related to SARS-CoV-2 (PIMS-TS) [1,2,3]. This condition showed overlapping clinical features with previously known diseases including Kawasaki Disease (KD), Toxic Shock Syndrome (TSS), and Macrophage Activation Syndrome (MAS). MIS-C is a severe complication of COVID-19 that often requires intensive care unit (ICU) admittance due to severe cardiac involvement and shock at onset, but also multiple organ failure that may develop in the disease course [2,4,5]. MIS-C usually starts 2–6 weeks after a benign SARS-CoV-2 infection. The prevalence of the condition has been estimated at about 45 cases per 100,000 (0.045%) infected children during the first pandemic wave, with lowering incidence through the following SARS-CoV-2-variants waves and a mortality rate below 2% [6]. Case definitions for this syndrome have been developed by the World Health Organization (WHO), the Centers for Disease Control and Prevention (CDC), the European Centre for Disease Prevention and Control (ECDC), and the Royal College of Pediatrics and Child Health (RCPCH) [7,8,9]. The diagnostic criteria are based on the presence of fever, clinical symptoms of different organs dysfunction, signs of systemic inflammation in laboratory tests (such as increased ferritin, C-Reactive Protein (CRP), Troponin T (TnT), Brain Natriuretic Peptide test (BNP or NT-proBNP)), and the association with SARS-CoV-2 infection in subjects 21-year-old or less. The CDC’s case definition is the most used of the four available, and probably includes a wider group of patients as it defines a smaller fever period as criteria (24 h). The condition appears to be more frequent in male children over five, even though some cases have been diagnosed in younger infants. Fever must be present in all patients, variably associated with gastrointestinal, respiratory, neurological, or renal signs and symptoms, and cardiac dysfunction. First-line therapy usually includes a single dose of intravenous immunoglobulin and high-dose glucocorticoids; additionally, anakinra has been used successfully in the case of critically ill or refractory MIS-C patients. Low-dose aspirin is usually started in all patients and continued for at least 4 weeks in those with coronary artery abnormalities (CAAs); anticoagulation with Enoxaparin is indicated in selected cases of documented thrombosis or with an ejection fraction below 35% [10,11].

The pathogenesis of MIS-C has been largely investigated. The disease is considered a post-infectious syndrome caused by the upregulation of different pro-inflammatory mediators, such as the monocyte-derived cytokines IL-1RA, IL-6, IL-10, IL-18, MCP1 (CCL2), and TNF, or the interferons (IFNs). It is still not clear if the hyperinflammatory state is induced by SARSCoV-2 acting as a superantigen, or by a mechanism of antibody-dependent enhancement.

There is now evidence that the SARS-CoV-2 spike protein may contain a superantigen-like motif, which can bind both MHC class I molecules and T cell receptors [12]. Case descriptions reported of lymphopenia in MIS-C patients, and the investigation of the immunological behavior of MIS-C patients has shown, in ~75% of patients, the polyclonal expansion of CD4+ and CD8+ T cells bearing the Vβ21.3 segment and TCR V11-2 skewing in association with HLA A*02, B*35 and C*04 [13]. All these findings point to a possible superantigen conveying the disease [14].

SARS-CoV-2 infection is characterized by extreme inter-individual clinical variability. In an effort to understand the underlying causes of this variability, molecular and genetic studies have revealed inborn errors of type I interferon (IFN) immunity and their phenocopies—autoantibodies against type I IFNs—collectively in at least 15% of cases of critical COVID-19 pneumonia in unvaccinated patients. Similarly, genetic studies have been conducted in MIS-C patients to identify monogenic inborn errors of immunity to SARS-CoV-2 that could explain the occurrence of the syndrome [15,16,17]. Recently, autosomal recessive deficiencies of *OAS1*, *OAS2*, or *RNASEL* have been identified in five unrelated children with MIS-C [18].

Therefore, MIS-C may represent a disease model of a hyperinflammatory condition secondary to a viral infection. The comprehension of the underlying immunological and genetic mechanisms may guide clinicians to better management and to target therapeutic options.

We conducted a longitudinal study to analyze a single-center cohort of MIS-C patients to characterize retrospectively their immune response during the acute phase of the disease and the outpatient long-term follow-up.

## 2. Materials and Methods

We conducted a longitudinal study at the Pediatric Department of ASST Spedali Civili of Brescia. We included all children and adolescents who were diagnosed with MIS-C, in accordance with the CDC definition [8] and our internal MIS-C diagnostic and therapeutic protocol, from April 2020 to June 2022. The CDC’s criteria considered the patient’s age defined as <21 years; fever defined as ≥38 °C, and present for at least 24 h; laboratory signs of inflammation; and the evidence of SARS-CoV-2 infection that could be current or even recent, as demonstrated by molecular biology detection (RT-PCR), positive serology, and antigenic test or COVID-19 exposure within the prior 4 weeks.

Patient care and research were conducted in compliance with the Declaration of Helsinki. Informed consent was acquired from patients’ parents, according to ASST Spedali Civili of Brescia protocols.

For all patients, we recorded demographics, length of hospital stays, need for pediatric intensive care unit (PICU), comorbidities, clinical presentation, and laboratory data that included full blood count and inflammatory markers: C-Reactive Protein (CRP), ferritin, Troponin T (TnT), and N-Terminal pro B-type Natriuretic Peptide (NT-proBNP). A cardiological evaluation was also carried out and included echocardiographic examination and electrocardiogram.

We considered as basal values the biochemical and blood count parameters that had been evaluated within 48 h from hospital admission and prior to the initiation of systemic therapy. These parameters had been monitored with daily frequency during the acute phase and at reduced frequency during the subacute phase, according to amelioration of the patients’ clinical conditions, until discharge.

A detailed multiparametric flow cytometry evaluation of immunological markers had been performed at baseline, and during the hospitalization in the acute phase of the disease and follow-up. Flow cytometry analyses were performed on fresh blood samples using four different multiparametric panels including appropriate mixtures of monoclonal antibodies (mAbs) according to manufacturer’s protocols. A standard TBNK panel was used on Becton Dickinson Trucount™ tubes to determine the percentages and counts of main lymphocyte populations (CD3+, CD4+, CD8+ T lymphocytes and their HLA-DR+ activated fraction, CD19+ B lymphocytes, and CD3-CD56/CD16+ NK cells). T-cell differentiation was assessed by using anti-GammaDelta TCR, CD45RA, CCR7, CD31 mAbs, on CD3+, CD4+, CD8+, and CD4-CD8- double negative cells. This strategy allowed for identifying CD4+CD45RA+CCR7+CD31+ Recent Thymic Emigrants (RTEs), and different maturation stages of T cells according to CD45RA and CCR7 expression: CD45RA+CCR7+ naïve, CD45RA+CCR7- central memory, CD45RA-CCR7- effector memory, and CD45RA+CCR7- terminal differentiated cells. Analyses of B-cell differentiation was assessed by using anti-CD19, CD20, IgM, IgD, CD38, CD27, CD21, and CD10 mAbs, which allowed for identifying CD19+CD20+ subsets such as Recent Bone Marrow Emigrants (RBE) (CD38++CD10+), naïve (CD38+IgD+IgM+CD27-), switched memory (CD38+/-IgD-IgM-CD27+), unswitched memory (CD38+/-IgD+IgM+CD27+), and CD38low-CD21low- B cells, along with CD38+++CD27+CD20- terminally differentiated B cells. Plasmacytoid (pDC) and myeloid (mDC) dendritic cells were identified as CD45dimCD4dimCD123+BDCA2+ cells and CD45dimCD4dimCD1c+CD14- cells, respectively, while basophils were gated as CD45dimCD4-CD123+BDCA2- cells. Expression of CD4 on monocytes was evaluated by considering both the percentage of CD4- cells and a Mean Fluorescence Intensity (MFI) ratio between CD4 expression on resting T helper cells and on monocytes, as previously described [19].

Multiparametric flow cytometry samples were acquired on FACSCanto II (BD Biosciences, Franklin Lakes, NJ, USA) flow cytometers, while data analyses were performed using FACSDiva 9.0 software (BD Biosciences).

For data analyses, we calculated and analyzed normalized values obtained by dividing patients’ values by the median of their own age-matched reference range (herein termed as normalization ratio). For this purpose, we used datasets previously obtained from pools of healthy donors at our laboratory. For the main lymphocyte subsets (total CD3+ T lymphocytes, CD3+CD4+ and CD3+CD8+ T lymphocytes, total CD19+ B lymphocytes, NK cells), this calculation was made considering data of absolute counts, while for T- and B-cell subsets considering their percentages with respect to their parental populations. Patients’ data were categorized into four longitudinal groups: (1) the first 5 days of hospitalization (16 determinations); (2) the second phase of hospitalization, the subacute phase, up to discharge (from day +6 to day +16, 15 determinations); (3) short-term follow-up at 20–60 days (21 determinations); (4) long-term follow-up at 100–414 days (31 determinations).

The database was formatted by using Microsoft Excel^®^ software (365 version) and later imported from IBM-SPSS^®^ software ver. 28.0.1 (IBM SPSS Inc., Chicago, IL, USA). Use of Stata^®^ software ver. 17.0 (Stata Corporation, College Station, TX, USA) was also considered for comparisons or implementations of test output. Normality of the distributions was assessed using the Kolmogorov–Smirnov test. Categorical variables were presented as frequencies or percentages and compared with the use of the chi-square test and Fisher’s exact test, as appropriate; associations of the cross-tabulations were verified using standardized adjusted residuals. Continuous variables were presented as means ±SD (in case of a normal distribution), or medians and min/max (in case of a skewed distribution) and compared with the use of Student’s T-test, Anova, or the Mann–Whitney test and Kruskal–Wallis test, and also with Wilcoxon and Friedman tests; correlations among variables were analyzed by the Pearson’s or Spearman’s rank correlation test. A two-sided α level of 0.05 was used for all tests. The authors had full access to the data and assume full responsibility for their integrity.

## 3. Results

### 3.1. Clinical Evaluation

Our cohort included 40 patients, 22 males and 18 females (M/F 55/45%). The median age at admission was 5.5 years (range 9 months to 17 years), with 24 patients being 5 years or older. The median duration from symptoms onset to hospital admission was 4 days and the hospitalization lasted a median of 10 days (range 2–27 days); 9/40 patients (22%) required PICU admittance: 5/9 patients (8/40, 20%) experienced shock, and 6/9 (6/40, 15%) needed inotropic support; 5/40 patients (12%) were given non-invasive ventilation. No patients required intubation and mechanical ventilation.

Fever was present in all patients. Rash was present in 21/40 patients (52%), limbs edema in 5/40 patients (12%), conjunctivitis in 10/40 (25%), and adenitis in 15/40 (37%); 28/40 patients (70%) had gastrointestinal symptoms (15/28 diarrhea, 4/28 nausea and vomiting, 7/28 abdominal pain, 3/28 terminal ileitis, 1/28 biliary sludge, 8/28 splenomegaly, 3/28 hepatomegaly, 3/28 abdominal multiple adenopathies), 17/40 patients (42%) showed respiratory symptoms (rhinitis, pharyngitis, cough), 6/40 (15%) neurological involvement (all with meningismus), 6/40 (15%) renal involvement (all with acute renal insufficiency), and 25/40 (62%) cardiovascular involvement.

Among patients with cardiovascular involvement, 10/25 patients (40%) developed coronary artery dilatation (Z score 2.39 ± 0.33, mean ± SD), 11/25 (44%) presented mild to moderate mitral valve regurgitation, 8/25 (32%) had left ventricular dysfunction, 4/25 (16%) had pericardial effusion, and 5/25 (20%) had signs of myocarditis, with mild diffuse hypokinesia at transthoracic echocardiography associated with elevated troponin T levels. Ventricular dysfunction normalized in all children during admission. None of the reported patients had congenital heart disease or preexisting cardiovascular disease. No major common preexisting comorbidities were recorded, except for a patient with chronic renal insufficiency in solitary kidney and neurogenic bladder.

Additionally, 5/40 patients (12%) were treated with sole IVIG at the conventional dose of 2 g/kg, 30/40 patients (75%) with IVIG and glucocorticoids (intravenously and/or orally), 4/40 patients (10%) received only glucocorticoids. A patient with myocarditis received non-steroidal anti-inflammatory drugs only.

Demographics and clinical symptoms are reported in Table 1.

### 3.2. Laboratory Parameters

#### 3.2.1. Inflammatory Markers

The total number of inflammatory markers measures comprised 178 determinations of CRP (days 1–21) (Figure 1); 107 determinations of ferritin (days 1–21); 82 determinations of TnT (days 1–12); and 61 determinations of NT-proBNP (days 1–14) (Appendix A).

The median value of CRP was 113 mg/L (reference value < 5 mg/L) at onset. Median CRP value, although not statistically significant, was higher in the four patients who were treated with glucocorticoids therapy only (177 mg/L), compared to the five patients treated with sole IVIG (106 mg/L) and most patients who received both therapies (117 mg/L). We did not observe significant differences among patients grouped according to age (median CRP of 106 mg/L in 16 patients < 5 years of age, compared to 136.5 mg/L in 22 patients > 5 years of age), nor according to severity (10 patients were considered as having a severe course of the disease because of PICU admission and/or the occurrence of cardiovascular shock had a median CRP value of 170 mg/L, compared to 106.5 mg/L for the group of non-severe patients).

Overall, the beginning of therapy corresponded to a significant decrease in CRP values progressively from day +1 (median 139.5 mg/dL) up to day +5 (median 52 mg/dL) (Figure 1A). Afterward, CRP values successively normalized: at the last determination at 0–3 days before discharge, the median value was 7.3 mg/L (Figure 1B).

At onset, the other biochemical inflammatory parameters, ferritin, troponin T, and NT-proBNP, had median values of 413 ng/L (12/23 patients out of the reference values of 413–2625 ng/L), 17 ng/mL (15/30 patients over 14 ng/mL), and 767 ng/L (13/20 patients over 300 ng/L), respectively. All these markers ameliorated at discharge (median values were 249 ng/L for ferritin, 5.5 ng/mL for troponin T, and 433 ng/L for NT-proBNP) (Figure 1). These inflammatory markers were negative during short- and long-term follow-up.

#### 3.2.2. White Blood Cells

Analyses of white blood cell counts were performed from hospital admission to discharge (214 determinations). At first determination, median total lymphocyte count of the whole cohort was 1430 cells/µL. Median values of patients < and >5 years were significantly different (2080 vs. 1190 cells/µL, *p* = 0.015). No differences were observed in more severe patients requiring PICU admittance and/or experiencing shock.

Concerning the normalized ratio, 34/39 patients (87%) displayed a total lymphocyte count below age-matched medians at presentation, that in 20 cases (51% of patients) fitted with a condition of lymphopenia, with counts below the 5th percentile of reference values. Total lymphocyte counts increased after the beginning of treatment and displayed a statistically significant amelioration from day +3 (Figure 1C). From day +6, signs of lymphocytosis emerged and persisted up to the day of discharge. At last determination before discharge, 34/40 patients (85%) displayed a total lymphocyte count over their median and 10/40 (25%) over their 95th percentile value.

During the whole follow-up period (from day +21 to day +414), 60 additional blood count determinations had been made. We observed that total lymphocyte counts showed a tendency to exceed the median during follow-up (41/60 determinations), although in only two cases total lymphocyte counts were above the age-matched 95th percentile. At follow-up, the Wilcoxon signed ranks test of normalized total lymphocyte counts showed significantly increased levels compared to first determination during admission, but also a significant decrease compared to the last blood counts before discharge (Figure 1D).

At admission, neutrophil counts displayed a high heterogeneity (median 8070 cells/μL, range 610–27,710 cells/µL), and neutrophilia was observed in 19/39 patients (49%) (Figure 1E). During the first 5 days of hospitalization, neutrophil counts did not show any significant difference (persistent neutrophilia in 8/18 patients with a median value of 8560 cells/µL). Before discharge, neutrophil counts decreased (median of the whole cohort 5045 cells/µL) (Figure 1F), and during follow-up neutrophil counts normalized in all patients (median 3770 cells/µL). Neutrophilia was more evident in more severe patients (median 10,290 vs. 7810 cells/μL, respectively), although without any statistically significance. In our cohort, MIS-C patients displayed a median of 400 monocytes/µL at onset (Figure 1G). No significant changes in monocyte count were observed in the first 5 days of hospitalization (median 470 cells/µL) but raised at discharge (median 1050 cells/µL, monocytosis in 13/43 patients (30%)) (Figure 1G,H). No significant differences in terms of monocyte counts were observed among different subgroups of patients.

As for adult patients affected by COVID-19 [20], monocytes of MIS-C patients displayed abnormal expression of surface markers in the acute and, partially, in the subacute phases of the disease, as measured in 26 patients of our cohort. At presentation, expression of CD4 was markedly reduced and associated with an increase in both T helper cells/monocytes ratio of MFIs and the percentage of CD4- monocytes (Figure 2A, and Appendix A). Firstly, we observed a median of 18% for CD4- monocytes (up to 55.3%). The amount of CD4- monocytes significantly dropped in the subacute phase (median 1.7%, *p* < 0.001) except for a single case that exhibited 26% negative cells. CD4- monocytes decreased further during short-term follow-up (median 0.25%, *p* = 0.008 compared to the subacute phase), and finally disappeared for all patients in long-term follow-up determinations (Figure 2A, and Appendix A). Of note, out of 15 patients who had been screened in the acute phase of the disease, the 3 patients who were admitted to PICU displayed a median T helper cells/monocyte MFI ratio of 49.7 (compared to 9.9 for the other 12 screened patients), with a percentage of CD4- monocytes equal or higher to the whole median of 18%.

At onset, dendritic cells and basophils counts were markedly deficient (median 0.4, 0.2, and 2.7 cells/mL for mDC, pDC, and basophils, respectively), and even undetectable in some of the 15 patients tested (Figure 2B–D). This deficit was particularly severe in 3 patients requiring intensive care compared to the other 12 patients (0.14 vs. 0.86 cells/μL for mDC, 0.14 vs. 0.38 cells/μL for pDC, and 1.8 vs. 3.5 cells/μL for basophils). In the subacute phase, cell counts started increasing (median of 10.4, 1.7, and 14.1 cells/µL, for mDC, pDC, and basophils, respectively), although pDC remained statistically deficient (*p* < 0.001), and normalized at long-term follow-up (median of 21.6, 35.4, and 48.3 for mDC, pDC, and basophils, respectively) (Figure 2B–D, Appendix A).

#### 3.2.3. T Cell Subsets and NK Cells

As described in the Methods section, flow cytometry data were normalized by dividing each patient’s value by the median of the corresponding age-matched reference range. Thus, compared to the reference medians, total CD3+ T lymphocyte counts and the counts of CD4+ and CD8+ subsets were significantly reduced during the first days of hospitalization (patients’ normalized medians ranged between 0.379 and 0.427, with *p* < 0.001 for all the three populations) (Figure 3A–C, and Appendix A). In the subacute phase, CD4+ and CD8+ T-cell subsets significantly increased (normalized median 1.447 and 1.553, respectively, with *p* < 0.001 comparing datasets of the first two longitudinal groups). During follow-up, CD4+ T cell counts progressively approached age-matched control medians (1.081 and 0.931 in the two follow-up groups, respectively, with *p* = 0.038 comparing long-term follow-up values to counts of the subacute phase) (Figure 3A,B). In contrast, normalized medians of CD8+ cells remained stably higher (1.538 and 1.437 at short- and long-term follow-up, respectively).

NK cells and GammaDelta+ T cells were evidently reduced during the first days after hospital admission (normalized median 0.298 and 0.201, respectively) (Figure 3D,E). Although both cell subsets significantly increased in the subacute phase, they remained below the normalized medians (0.911 and 0.667, with *p* < 0.001 and *p* = 0.017, respectively, compared to values at admission) (Appendix A). During follow-up, their median values further increased and became not significantly different from normalized medians (Figure 3D,E).

CD4+ and CD8+ T cells were analyzed in their activation and maturation profile as well. The evaluation of the fractions of HLA-DR+ activated T cells highlighted the presence of a raised activation during the first 5 days of hospitalization (normalized median 1.360 and 2.269 for CD4+ and CD8+ cells, respectively, with *p* = 0.047 for CD8+ cells) (Figure 3F,G). The percentage of HLA-DR-expressing T lymphocytes dropped in the subacute phase (normalized median 1.068 and 0.907 for CD4+ and CD8+ cells, respectively). At follow-up, the fraction of activated T cells decreased further, significantly differing from values at presentation for both CD4+ and CD8+ T lymphocytes (Figure 3F,G, Appendix A).

The distribution of the CD4+ cells subsets corresponding to the main maturation stages that are detectable in the peripheral blood did not show significant variations (Appendix A). Normalized medians of naïve, recent thymic emigrants (RTE), central and effector memory, and regulatory CD4+ T cells showed only slight fluctuations. RTE basically reflected longitudinal changes observed for total CD4+ T cells, except for a softer increase after the fifth day of hospitalization (Figure 3H, Appendix A). Conversely, terminally differentiated CD4+ T cells markedly increased (normalized median 1.026 at presentation, 1.995 and 1.892 at short- and long-term follow-up), although without any statistical significance (Appendix A).

Conversely to CD4+ T cells, the cytotoxic counterpart of T lymphocytes displayed significant changes in the distribution of cell subsets after hospital discharge. Specifically, during the acute and subacute phases, CD8+ naïve T cells remained significantly above the normalized median (1.115 and 1.163, respectively, with *p* < 0.001), while we observed values below the normalized median for both the CD8+ effector T cells (0.571 and 0.599, respectively) and CD8+ terminally differentiated T cells (0.841 and 0.762, respectively, with *p* < 0.001) (Figure 3I–K). Of note, normalized medians for both CD8+ effector and terminally differentiated T cells were particularly reduced in the three most severe analyzed patients (0.488 and 0.369, respectively, vs. 0.603 and 0.956 for patients who did not present serious clinical course). The distribution of CD8+ T subsets observed during hospitalization was reversed at follow-up: naïve cell counts fell below their normalized median (0.733 at long-term follow-up, with *p* < 0.001); in contrast, during short- and long-term follow-up, we observed a significant increase in CD8+ for both effector (0.883 and 0.931, respectively) and terminally differentiated T cells (1.438 and 1.517, respectively) (Figure 3I–K, Appendix A).

#### 3.2.4. B Cell Subsets

Conversely to the other lymphocyte subsets, we did not observe B cell lymphopenia during the first 5 days of hospitalization (normalized median 0.928) (Figure 3A), although a reduced normalized median (0.693) was present in the three most severe patients. In the subacute phase, total B cell count significantly increased (normalized median 2.261, *p* < 0.001). During follow-up, B cell count dropped and constantly remained slightly below normal medians (Figure 4A, and Appendix A).

However, we observed some significant variations in the longitudinal distribution of B cell subsets. RBE subsets were normal at presentation (normalized median 1.009), but significantly dropped in the subacute phase (0.572, *p* = 0.014), in correspondence with the increase in total B cell count. They decreased further during the first weeks after discharge (0.435), and finally returned close to expected values at long-term follow-up (1.115, *p* < 0.001). Interestingly, the RBE subset decrease was more evident in patients experiencing the most severe course of the disease (normalized median 0.251 vs. 0.560) (Figure 4B, and Appendix A).

Concerning the RBE subset, naïve B cells initially equalized expected values (normalized median 0.996), then peaked during the B lymphocytosis phase (1.105), and finally slightly declined at long-term follow-up (0.857, with *p* = 0.006 compared to the subacute phase) (Figure 4C, and Appendix A).

Memory B cells were mostly below the normalized median at presentation (0.696 and 0.744 for switched and unswitched B cells, respectively). In correspondence to B cells lymphocytosis, we observed a rapid expansion of switched memory B cells (0.985, *p* = 0.008) (Figure 4D, and Appendix A), mainly sustained by the most severe patients (normalized median 2.162 in five patients). Then, both memory switched and unswitched B subsets peaked a few weeks after hospital discharge (1.214 and 1.087, with *p* = 0.001 and *p* = 0.003 compared to onset, respectively), and finally settled close to the expected values at long-term follow-up (0.995 and 0.864, respectively) (Figure 4D,E, and Appendix A). Similarly, CD21low atypical B cells were initially reduced (0.603), but then progressively increased and peaked at long-term follow-up (1.170, *p* = 0.010 comparing first and last measurements) (Figure 4F, and Appendix A).

Terminally differentiated B cells, which were significantly above the normalized median at onset (2.141, *p* = 0.001), rapidly fell in the subacute phase (0.456, *p* < 0.001), and finally approached expected values during short- and long-term follow-up (0.800 and 0.758, respectively) (Figure 4G, and Appendix A).

## 4. Discussion

Multisystem inflammatory syndrome in children is a rare but severe complication of COVID-19, known since 2020, that results from a hyper-reactive dysregulated immune response. From April 2020 to June 2022, at the Department of Pediatrics of the ASST Spedali Civili of Brescia, 40 patients were admitted for MIS-C.

All patients were treated according to our standard protocol [3], based on an updated review of the literature and a multidisciplinary expert consensus. In most of the patients (75%), the cardinal treatment consisted of IVIG at 2 g/kg and corticosteroids (intravenous methylprednisolone 2 mg/kg, followed by tapering with oral prednisone). A small group of five very young patients with a milder clinical picture and milder abnormalities in the biochemical and immunological parameters (CRP and lymphocyte/neutrophil counts) rapidly improved with sole IVIG, while four other patients were treated only with corticosteroids.

MIS-C patients exhibited increased inflammatory markers at presentation, with CRP as the most extensively measured marker. Although slightly heterogeneous, we did not observe significantly different CRP values comparing patients’ subgroups at first measurement. After the beginning of treatment, CRP displayed a day-by-day significant decrease that reduced the initial value by half on day +5 and normalized before discharge in one-third of patients (13/39) and in all patients at short-term follow-up.

Neutrophil activation is a distinct feature of Kawasaki disease and has already been reported in patients with MIS-C [21,22]. In our cohort, neutrophilia affected half of the patients at presentation and was more evident in more severe cases (median 10,290 vs. 7810 cells/μL, respectively). During treatment, neutrophil counts tended to normalize, with only six patients (15%) presenting with neutrophilia at discharge.

Conversely to neutrophils, monocyte count was initially normal in 78% of the patients, then increased in the subacute phase of hospitalization (median values 400 vs. 1050 cells/μL). Although the total number of monocytes has been reported to be normal in MIS-C, there is also evidence of the active recruitment of this cell population in some MIS-C patients, as shown by Gruber et al. [21]. A shift in monocyte counts paralleled with normalization of the expression of surface-specific markers. We focused on the expression of CD4, because of its ease to evaluate in very common flow cytometric panels and its strict temporal correlation with the amelioration of CRP values and lymphocyte counts; in fact, the median of CD4- monocytes dropped from 18% in the acute phase of the disease to 1.7% from day +5 onward, before slightly disappearing at the last follow-up determinations (Figure 5A). Moreover, in our experience, CD4 expression may correlate with clinical conditions at presentation, as suggested by its increase in three of our most severe patients. A similar trend was observed also in dendritic cells and basophils counts. Consistent with literature and our previous results [23], we confirmed a profound depletion of dendritic cells at admission, which was more marked in the most severe patients and that we herein reported also for basophils. The normalization of these cell subsets’ counts was slower compared to the rapid improvement we conversely observed for lymphocyte subsets, and completed only at long-term follow-up (Figure 5B). In particular, median pDC count remained below 1/20 of long-term follow-up values for the entire duration of the hospital stay.

It is recognized that lymphopenia represents the most evident immunological hallmark of MIS-C. Considering normalized values, we reported that 51% of patients displayed lymphopenia at onset, while 87% presented with total lymphocyte counts below the median of the reference value for their age. Interestingly, the prevalence of lymphocyte counts below the normalized median disappeared after 5 days of hospitalization. Conversely to what expected, after day +5 total lymphocyte counts kept on increasing. Thus, at discharge 25% of patients presented with lymphocytosis, while 85% of them displayed lymphocyte counts above the normalized median. The lymphocytosis phase was temporary and at short-term follow-up total lymphocyte counts were significantly lower and much closer to the normal range, despite a tendency to remain above the normalized median value. No significant differences in total lymphocyte counts were observed between patients’ subgroups according to age, treatment, and severity. More severe patients did not display a higher degree of lymphopenia at presentation, or a different incidence of lymphocytosis in the subacute phase of the disease. The acute phase of the disease was characterized by a significant reduction in T- and NK-cells, with GammaDelta+ cells as the T subset most dramatically affected by the disease (normalized median 0.201 vs. 0.379 of the whole T-cell compartment), followed by NK cells (normalized median 0.298). Our finding confirms the reduction in these populations in acute MIS-C patients already observed by Gruber et al. [21].

CD4+ and CD8+ T cells were also significantly reduced at onset (normalized median 0.399 and 0.427, respectively). As already reported by Carter et al. [22], both subsets displayed a significant increase in activated cells compared to normalized medians. In contrast, while CD4+ maturation profile was consistent with the expected distribution in healthy condition, CD8+ cells showed a reduction in more differentiated subsets, especially in the three patients requiring admission to the intensive care unit (Figure 5C).

Interestingly, the same three patients displayed a significant reduction in total B cell counts at onset, a condition limited to few cases in our cohort. At the level of B cells, the inflammatory status in the acute phase of the disease matched with a general expansion of terminally differentiated cells, counterbalanced by a reduction in the memory compartment (Figure 5D).

The subacute phase of the disease was associated with marked changes in lymphocyte counts. We observed a four-fold increase in CD4+, CD8+, and GammaDelta+ T-cell subsets within a few days (0.399–1.447, 0.427–1.553, and 0.201–0.911, respectively), while B lymphocytes markedly expanded to a maximum seven-fold increase. Instead, the increase in NK cell count was less evident (0.298–0.667) and remained below the normalized median. In this phase of the disease, T-cell subsets showed a significant decrease in HLA-DR+ activated cells, but only minimal changes in the distribution of subsets associated with different maturation stages. We also observed rapid changes in the level of B-cell subsets. RBE cells and terminally differentiated cells dropped. This last shift, together with the reduced activation status of T cells, may correspond to the switching-off of the hyperactivation phase of the disease. Naïve B cells displayed their highest peak in the subacute phase, indicating that they were the greatest contributors to B-cell count increase, together with the rise in the proportion of switched memory cells, especially observed in the more severe patients. In contrast, a T-cell increase may be ascribable to a general rise in all the subsets, perhaps due to a peripheral expansion or, more reasonably, to a mobilization of T lymphocytes from other districts.

At follow-up, the apparent quiet trend toward normal counts of most of the main lymphocyte populations hid a constant evolution in both B- and CD8+ T-cell subset distribution (Figure 5C,D). In fact, at the level of the B-cell compartment, during the short-term follow-up, the depletion in the RBE subset became more pronounced (normalized median of 0.435), while median values for both switched and unswitched memory B cells displayed their highest points. These alterations progressively attenuated during follow-up. Of note, we observed a constant increase in atypical CD21low during follow-up, which in some cases reached more than a three-fold increase compared to reference medians in the last determinations.

CD8+ T cells constituted the lymphocyte subset more consistently altered at follow-up. In fact, their counts remained elevated even at long-term follow-up (normalized median 1.64 at least 300 days after hospital admission). Persistence of elevated CD8+ T cell counts was mainly sustained by terminally differentiated cells, which remained significantly above the normalized median even at longer time points (normalized median 1.61 at least 300 days after hospital admission). In contrast, percentages of naïve cells were significantly lower compared to normalized medians.

Of note, an even more pronounced long-term increase in terminally differentiated cells could also be observed in CD4+ T lymphocytes (normalized median of 2.65 at least 300 days after hospital admission), although this population has much less relevance than its counterpart at the level of CD8+ cells.

All the observations reported above represent an effort to depict a clear picture of the longitudinal changes in the immunological cell subset in a relatively small group of patients with MIS-C treated in our center according to standard criteria. Moreover, the distribution observed among the different cell subsets suggests the involvement of both innate and adaptive immunities in the pathogenesis of MIS-C. The perturbations described at presentation clearly resemble those immunological characteristics observed in acute COVID-19 adult patients [19,20]. Lymphopenia and neutrophilia, together with an increase in biochemical inflammatory markers and lymphocyte-activated subsets, are among the most known hallmarks of the disease. Additionally, the profound depletion of dendritic cells and the marked alteration in monocyte surface markers, particularly dramatic in MIS-C patients displaying the most severe course of the disease, may correlate with severe COVID-19 in adults. Of note, from the lymphocyte point of view, B-cell lymphopenia, and a higher reduction in terminally differentiated CD8+ T cells, characterized the most severe cases in the acute phase of MIS-C.

While the immunological changes observed in the first days after the initiation of the pharmacological treatment, such as increase in lymphocyte counts, drop of activated subsets, and bone marrow release of newly generated B cells, may depend on the immunomodulatory treatment on the acquired immune system, the latter abnormalities, such as long-term persistence of terminally differentiated T cells and increase in atypical CD21low and memory B cells, can be classed as inheritance of the inflammatory response (Figure 5C,D).

Limitations of the study include the limited sample size that impairs the strength of the analysis, and the retrospective nature of the immunological evaluation that caused missing data during hospitalization and follow-up. Moreover, most MIS-C patients were treated with IVIG alone or in association with glucocorticoids early in the course of the disease, and it was difficult to discriminate the role of the medical therapies on the evolution of the distribution of lymphocyte subsets. Nevertheless, our data may contribute to exploring the immune response of hyperinflammatory complications that may occur after viral triggers.

## 5. Conclusions

The assessment of the immune response in MIS-C may have implications for clinical practice. In fact, the characterization of the lymphocyte subsets may help the clinician to recognize and monitor the different phases of the disease and to recommend prompt treatments that may limit the progression of the disease and correct the hyperinflammation response. If confirmed on larger cohort, our finding that some immune-cell abnormalities (absence of dendritic cells, prevalence of CD4- monocytes, lymphopenia with skewing toward CD8+ naïve and activated T cells, and reduction in B cells) correlate with disease prognosis could be of great help when tailoring treatment in the single patient with MIS-C. Finally, we demonstrate the importance of the long-term follow-up to monitor the immune remission of the disease other than clinical remission after hospital discharge, showing that the complete normalization of lymphocyte distribution may require several months after the disease.

## Figures and Tables

**Figure 1 children-10-01069-f001:**
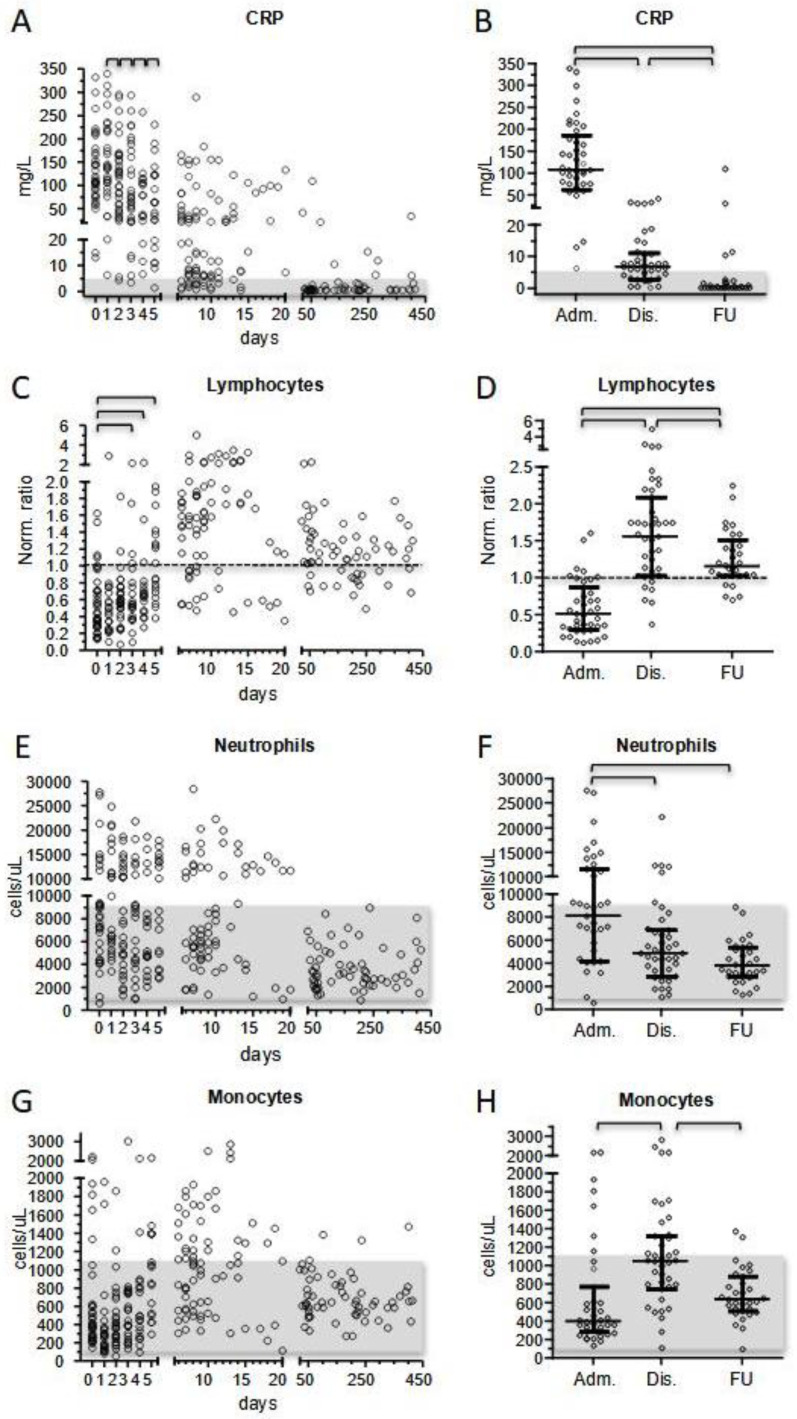
Patients’ biochemical and immunological parameters presented as longitudinal data (0–414 day from hospitalization) (left panels), and as comparison among data at hospital admission (Adm.), hospital discharge (Dis.), and follow-up (FU) (right panels). (**A**,**B**): C-reactive protein; (**C**,**D**): lymphocyte counts calculated as normalized ratios; (**E**,**F**): neutrophil absolute counts; (**G**,**H**): monocyte absolute counts. Bars of the scatter dot plots of panels (**B**,**D**,**F**,**H**) represent medians and interquartile range. Day-by-day analysis during hospitalization was performed by the Wilcoxon signed ranks test, while statistical analyses between groups of panels (**B**,**D**,**F**,**H**) were performed by the Mann–Whitney test. Horizontal segments indicate when *p* values were below 0.05.

**Figure 2 children-10-01069-f002:**
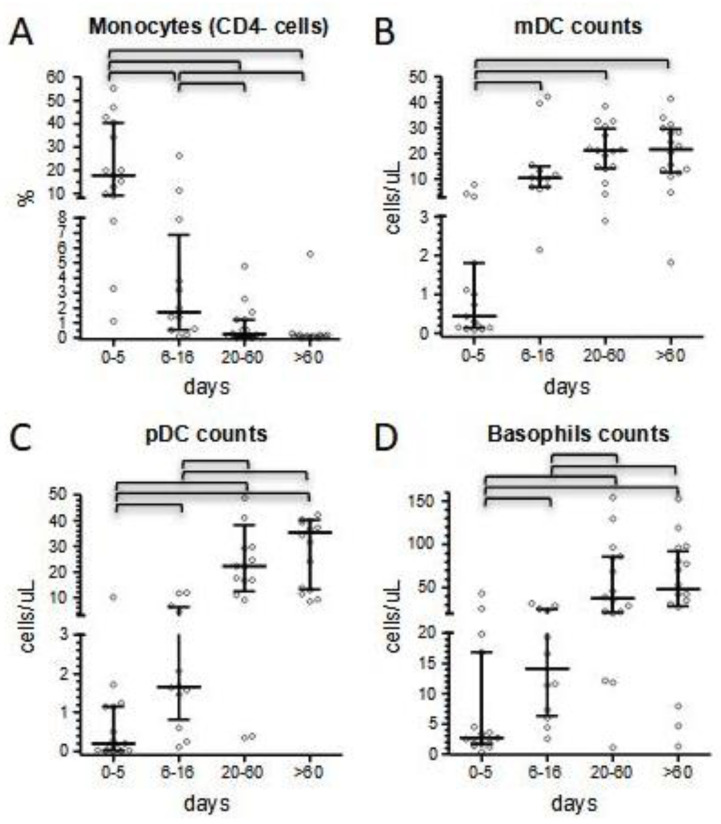
Comparison of patients’ immunological parameters at four different time points according to days after hospital admission (0–5 and 6–16 days during hospitalization; 20–60 and >60 days at short- and long-term follow-up, respectively). (**A**): percentage of CD4- monocytes; (**B**) myeloid dendritic cells absolute counts; (**C**) plasmacytoid dendritic cells absolute counts; (**D**) basophils absolute counts. Bars of the scatter dot plots represent medians and interquartile range. Statistical analyses between groups were performed by the Mann–Whitney test. Horizontal bars indicate where *p* values were below 0.05.

**Figure 3 children-10-01069-f003:**
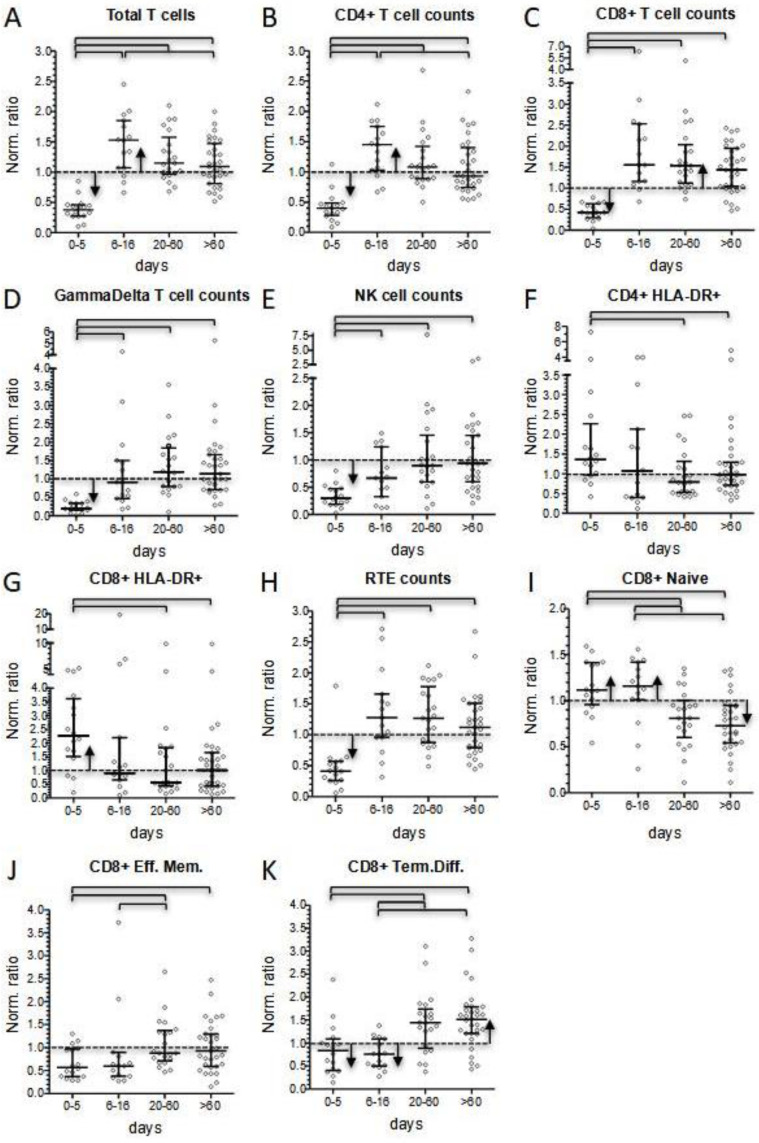
Comparison of patients’ lymphocyte subsets values, calculated as normalized ratios, at four different time points according to days after hospital admission (0–5 and 6–16 days during hospitalization; 20–60 and >60 days at short and long-term follow-up, respectively). (**A**): Total T lymphocyte counts; (**B**) CD4+ T lymphocyte counts; (**C**) CD8+ T lymphocyte counts; (**D**) GammaDelta+ T lymphocyte counts; (**E**) NK cell counts; (**F**,**G**) percentage of activated CD4+ and CD8+ cells; (**H**) Recent Thymic Emigrants (RTE) counts; (**I**–**K**) percentage of CD8+ cell subsets: naïve, effector, and terminally differentiated cells. Bars of the scatter dot plots represent medians and interquartile range. Statistical analyses between data for different time points were performed by the Mann–Whitney test. Horizontal bars among two groups indicate when *p* values were below 0.05. Pearson’s chi square test was performed to determine if data for each time point was significantly (*p* < 0.05) lower (down arrows) or higher (up arrows) from the normalized median (=1).

**Figure 4 children-10-01069-f004:**
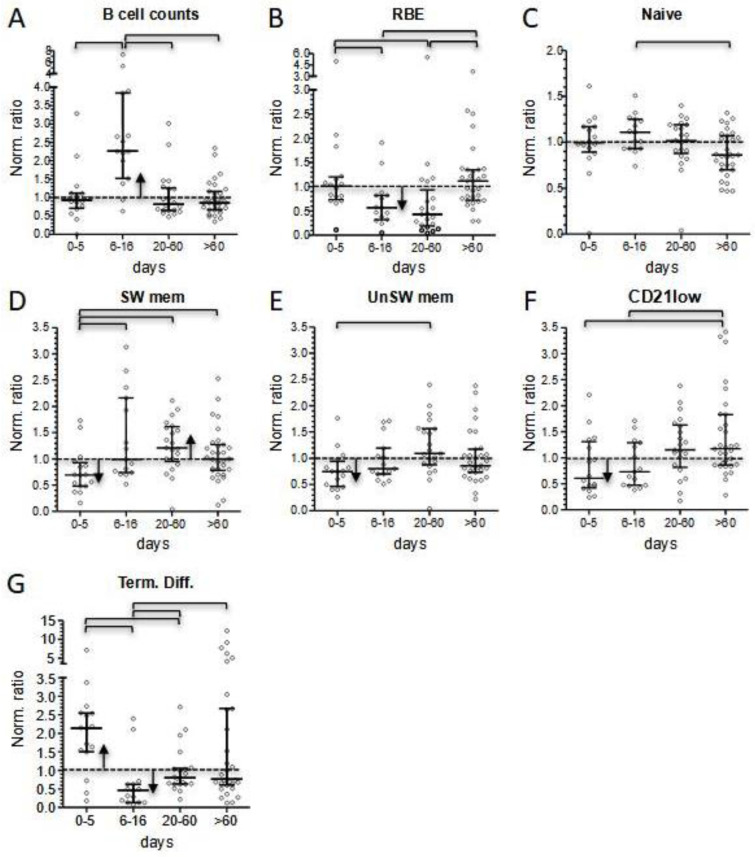
Comparison of patients’ B lymphocyte subset values, calculated as normalized ratios, at four different time points according to days after hospital admission (0–5 and 6–16 days during hospitalization; 20–60 and >60 days at short- and long-term follow-up, respectively). (**A**): Total B lymphocyte counts. Percentages of different B cell subsets: (**B**) recent bone marrow emigrants (RBE); (**C**) naïve; (**D**) switched memory; (**E**) IgM+ unswitched memory; (**F**) CD21low cells; (**G**) terminally differentiated. Bars of the scatter dot plots represent medians and interquartile range. Statistical analyses between data for different time points were performed by the Mann-Whitney test. Horizontal bars among two groups indicate when *p* values were below 0.05. Pearson’s chi square test was performed to determine if data for each time point was significantly (*p* < 0.05) lower (down arrows) or higher (up arrows) from the normalized median (=1).

**Figure 5 children-10-01069-f005:**
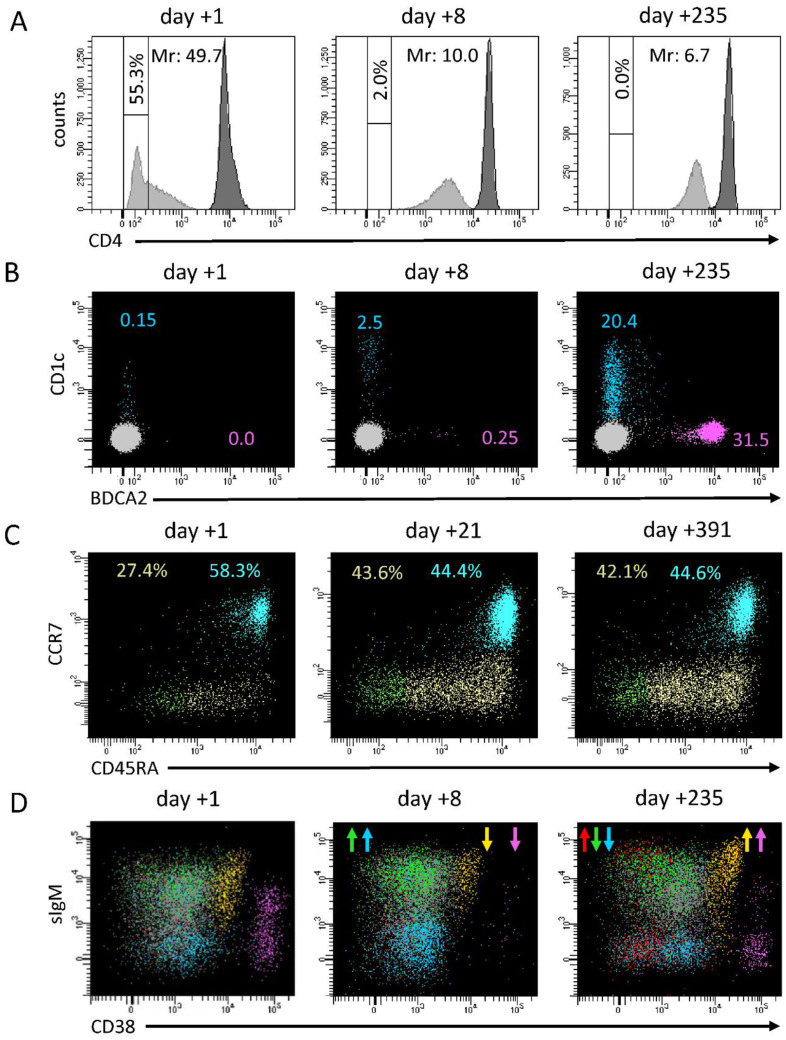
Representative panel of longitudinal changes in immunological subsets in two MIS-C patients analyzed in three different phases of the disease. (**A**) CD4 expression on monocytes both measured as percentage of CD4- cells and MFI ratio (Mr) between T helper cells (gray histogram) and monocytes (white histogram); (**B**) pDC (pink) and mDC (blue) evaluation, according to BDCA2 and CD1c expression among CD45+CD4+CD14- gated cells; (**C**) evaluation of CD8+ T-cell maturation stages: naïve (blue), central memory (orange), effector memory (green), terminally differentiated (yellow); (**D**) evaluation of B cell subsets: RBE (orange), naïve (gray), switched memory (blue), unswitched memory (green), CD21low (red), terminally differentiated (pink). Arrows indicate when a subset significantly increased or decreased compared to the previous determination.

**Table 1 children-10-01069-t001:** Clinical characteristics of MIS-C patients.

Demographics	MIS-C (*n* = 40)
Age (median, years)	5.5
Sex (M/F)	22/18 (55/45%)
Illness day of hospital admission	4
Length of hospital stay (days)	10
Need of PICU	9 (22%)
Main clinical manifestations	
Cardiovascular shock	8 (20%)
Coronary artery dilatation	10 (25%)
Left ventricular dysfunction	8 (20%)
Pericardial effusion	4 (10%)
Myocarditis	5 (12%)
Neurological involvement (meningismus)	6 (15%)
Gastrointestinal manifestations (nausea, vomiting, diarrhea, abdominal pain, hepatosplenomegaly, adenopathy, terminal ileitis)	28 (70%)
Respiratory symptoms (rhinitis, pharyngitis, cough)	17 (42%)
Treatment	
Intravenous immunoglobulins (group 1)	5 (12%)
IVIG+steroids (group 2)	30 (75%)
Steroids (group 3)	4 (10%)

PICU, Pediatric Intensive Care Unit; IVIG, Intravenous Immunoglobulin.

## Data Availability

Data are available at 10.5281/zenodo.7899859.

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
