# Peer review of "Longitudinal Characterization of Immune Response in a Cohort of Children Hospitalized with Multisystem Inflammatory Syndrome"

_children, 2023, doi:10.3390/children10061069_

Round 1

Reviewer 1 Report

Thank you for the opportunity to review this manuscript detailing an interesting analysis of peripheral blood immunological cell subsets in 40 children and adolescents with MISC. Your results are notable, well organized, and meticulously reported. Your discussion adds greatly to the manuscript, in several instances helping to explain your observations and put them in context, and I agree with your conclusions.

Please clarify the following:

The first sentence under Materials and Methods, and the Study Limitations paragraph of the Discussion, refer to this as a retrospective study design. However, in paragraph 5 under Materials and Methods you state that "a detailed multiparametric flow-cytometry evaluation of immunological markers was performed at baseline, during the hospitalization in the acute phase of the disease and during follow-up. Flow cytometry analyses were performed on fresh blood samples using four different multiparametric panels including appropriate mixtures of monoclonal antibodies (mAbs) according to manufacturer’s protocols." Please clarify how this was done retrospectively; you appear to be describing a prospective study method.

None

Author Response

We thank the reviewer for the observation. We specified in the Introduction section that this is a longitudinal study as we evaluated retrospectively the immunological parameters during hospitalization and follow-up. We detailed in the different sections of the manuscript the type of the study, accordingly.

Reviewer 2 Report

1. Please state the aim of this study in the last paragraph of introduction

2. in line 102, authors mentioned that this is a retrospective study. What type of retrospective study do this author used? is it a cohort study? longitudinal study? or case control study?

3. In line 104, authors mentioned that they used a CDC definition for MIS-C criteria. please include the reference in this paragraph.

4. why did authors did not use a multivariate/multivariable analysis? because there is differences in demographic and initial clinical characteristics. 

5. please create a dedicated section for conclusion

please check for your grammar because there are several grammar problems.

Round 2

Reviewer 1 Report

Thank you very much for addressing my comments. I agree with all of your responses and changes to the manuscript. Best of luck with your manuscript.

Thank you for your changes.

Author Response

We thank the Reviewer for all the comments and the revision. We tried to make minor editing, as suggested.

Reviewer 2 Report

The manuscript now is ready to be accepted

Author Response

We thank the Reviewer for the revision.